# The Latest Research on RT-QuIC Assays—A Literature Review

**DOI:** 10.3390/pathogens10030305

**Published:** 2021-03-05

**Authors:** Thi-Thu-Trang Dong, Katsuya Satoh

**Affiliations:** Department of Locomotive Rehabilitation Science, Nagasaki University Graduate School of Biomedical Sciences, 1-7-1 Sakamoto, Nagasaki 852-8501, Japan; trangneuro@gmail.com

**Keywords:** RT-QuIC assay, prion diseases, diagnosis, synucleinopathy, tauopathy

## Abstract

The misfolding of proteins such as the prion protein, α-synuclein, and tau represents a key initiating event for pathogenesis of most common neurodegenerative disorders, and its presence correlates with infectivity. To date, the diagnosis of these disorders mainly relied on the recognition of clinical symptoms when neurodegeneration was already at an advanced phase. In recent years, several efforts have been made to develop new diagnostic tools for the early diagnosis of prion diseases. The real-time quaking-induced conversion (RT–QuIC) assay, an in vitro assay that can indirectly detect very low amounts of PrP^Sc^ aggregates, has provided a very promising tool to improve the early diagnosis of human prion diseases. Over the decade since RT–QuIC was introduced, the diagnosis of not only prion diseases but also synucleinopathies and tauopathies has greatly improved. Therefore, in our study, we summarize the current trends and knowledge of RT–QuIC assays, as well as discuss the diagnosis of neurodegenerative diseases using RT–QuIC assays, which have been updated in recent years.

## 1. Introduction

Neurodegenerative diseases are characterized by the accumulation of disease-related misfolded proteins. Protein misfolding diseases such as prion diseases, Alzheimer’s disease (AD), Parkinson’s disease (PD), and related misfolded prion protein (PrP), tauopathies, and synucleinopathies represent a major disease burden on society.

Prion diseases, also known as transmissible spongiform encephalopathies (TSEs), are a group of progressive neurodegenerative disorders. Human prion diseases (HBD) include Creutzfeldt-Jakob disease (CJD), fatal insomnia, Gerstmann-Sträussler-Scheinker syndrome, kuru, and variably protease-sensitive prionopath. The hallmark event in prion disorders associated with the deposition of misfolded forms of the prion protein (PrP^Sc^). It is accepted that PrP^Sc^ formation is the key initiating event in prion disease pathogenesis. As such, PrP^Sc^ is also the most widely used diagnostic surrogate marker for affected individuals and animals.

Lewy bodies and Lewy neurites were defined as pathologic hallmarks of Parkinson’s disease over a century ago. For 20 years, α-synuclein was found to be the primary component of these inclusions. Emerging evidence suggests that α-synuclein pathology propagates across interconnected networks throughout the nervous system in a prion-like manner [1]. 

Among dementias, AD is the most common, with a worldwide prevalence of approximately 35 million people in 2010 [2,3], increasing to 50 million people in 2020 [4]. Total payments in 2020 for health care, long-term care and hospice services for people age 65 and older with dementia are estimated to be $305 billion [5]. AD is commonly thought to be a secondary tauopathy, and it has a pathogenic mechanism similar to that of prion disease [6]. The key neuropathological feature is the accumulation of tau proteins in the form of self-seeding filaments or sub-filamentous deposits [7].

Accurate diagnosis of neurodegenerative diseases is difficult when the patient is still alive, especially at the preclinical stage. Currently, there is no specific treatment, especially the late stage of the disease, but if the disease is detected in an early stage, we will have appropriate supportive treatment methods. One of the amplification techniques is real-time quaking-induced conversion (RT–QuIC) assays which can detect a small of prion seeding activity to exponentially amplify from various biopsies with high sensitivity and specificity. 

After more than a decade of the advent and development of the RT–QuIC assays [8,9], it has become a useful tool in the laboratory to aid in the clinical diagnosing of neurodegenerative disorders. To date, RT–QuIC assays have been developing to detect protein seeding activity for prion diseases in animals as well as in humans, α-synuclein in the synucleinopathies such as Parkinson disease (PD), multiple system atrophy (MSA), dementia with Lewy bodies (DLB) as well as tau aggregation in Alzheimer’s disease (AD), chronic trauma to encephalopathy (CTE), Pick disease (PiD).

In our review, we summarize the trends in the development of RT–QuIC assays in recent years, applying RT–QuIC assays not only for the diagnosis of prion disease but also for other neurodegenerative diseases.

## 2. The Advantages of RT–QuIC Assays

Resistance to proteinase K (PK) digestion is considered a definition of the transmissible isoform of the prion protein (PrP^Sc^). Thus, the gold standard of diagnosis for prion disease is detected proteinase K-resistant prion protein (PrP^Sc^). Like amyloid fibers found in Alzheimer’s and other amyloid-related disorders or tau aggregation in tauopathies, PrP^Sc^ is a variable polymer that can aggregate large fiber. The immune assay cannot distinguish between PrP and PrP^Sc^ because the distinguishing antibodies are not available. Western blotting (WB) can provide valuable information about the biochemical properties of PrP^Sc^, but its detection range is narrow and unsuitable for evaluating the decontamination of prion seeds [10]. For the evaluation of prion decontamination, bioassays are often used, but it takes at least 1 year to quantify infectivity. Similarly, quantitation of infectivity of tissue from a patient with human prion disease can also be achieved by animal bioassays using humanized mice [11,12]; however, these mice have different susceptibilities to human prion strains [13,14], and the assays are still highly time-consuming and costly. 

The protein misfolding cyclic amplification assay (PMCA) and RT–QuIC are the two most commonly used PrP^Sc^ amplification techniques. Although these two methods have the same basic methodology, they also have differences in their amplification and detection.

PMCA was first described in 2001 [15], in which brain homogenates [16] or cell extracts [17,18] were the source of PrP^C^. These methods could successfully diagnose variant Creutzfeldt–Jakob disease (vCJD) using samples from the brain and peripheral tissues such as the spleen and tonsil [19] and biological fluids such as blood [20] and urine [21]. PMCA can be applied in many fields and has a great tool of great significance for researching and diagnosing. This technique is used to understand the biology of unique prion, to detect ultrasensitive prion, and to screen for molecules that interfere with the formation of prion. Furthermore, the prions produced by PMCA can be infected in animals of various species of wild-type. This was supported by the fact that the agent of infecting in TSEs is composed exclusively of protein. Previous studies have also shown the infection material was created by PMCA using brain extracts, cell lysates, high purity PrP^C^, and even recombinant PrP produced in bacteria [22]. 

However, the PMCA method only detects the prion seeding activity in the variant in CJD. About 10 years later, a new technique was introduced, the “quaking-induced conversion” (QuIC), which improves the speed and practicality by using recombinant bacterial-expressed PrPC (rPrPc) and by replacing shake for sonication [9]. Moreover, although RT–QuIC is as sensitive as PMCA in the detection of as little as 1 fg PrP^Sc^ in the brains of sporadic CJD (sCJD) patients [9,23], PMCA requires brain homogenate, the sonication setup is much more difficult and expensive to maintain, and it needs PK digestion of samples and Western blotting for the detection of resistant PrP. These drawbacks make this technique highly unsuitable for the diagnosis of TSEs in hospitals or other medical centers. In addition, the RT–QuIC reaction products are not contagious and therefore do not require specialized containment laboratories. Finally, RT–QuIC, besides being low-cost and a multiwall Thioflavin T (ThT) detection form, is better adapted for use as a high-throughput diagnostic assay than PMCA.

## 3. The Development of RT–QuIC Assays and Future

RT–QuIC detects limited amounts of abnormal proteins. It is highly sensitive and specific in various specimens in the central and peripheral nervous systems but is not successful in samples containing blood, including whole blood, blood plasma, or blood-contaminated tissues. Previous studies have suggested that red blood cells in the Cerebrospinal fluid (CSF) sample inhibit the RT–QuIC response [24,25]. A cut-off value of < 1250 × 106/L has been recommended [25]. In addition, RT–QuIC has one of the diagnostic criteria sCJD. It is especially valuable in genetic diseases such as GSS and FFI that other CSF tests are negative. However, it has been inhibited in vCJD. 

Therefore, an adaptation of RT–QuIC, called e-QuIC, was developed [26]. The study showed that the eQuIC assay can detect up to 10^14^-fold dilutions and contains less than ~1 ag of PrP^Res^ vCJD brain homogenates in human plasma, so it was around 10^4^ times more sensitive than in the previous study. However, the assay was difficult to standardize and less reproducible than the standard RT–QuIC (given that eQuIC is significantly more sensitive than the RT–QuIC). 

Although there have been significant advances in recent years, the sensitivity and speed of first-generation RT–QuIC assays need to improve. One major advance has been developing a second-generation RT–QuIC assay using the N-terminally truncated Syrian hamster rPrP (SHarPrP (90–231)), comprising 0.002% sodium dodecyl sulfate (SDS) in the buffer solution and using a reaction temperature increased to 55 °C [27]. The initial evaluation of the IQ-CSF assay indicated greater analytical and diagnostic sensitivity and markedly shorter testing times [27,28,29,30]. Given the huge diagnostic and analytic potential of this technique, future studies should aim to develop a type-specific RT–QuIC to discriminate the molecular subtypes of sCJD cases in vitro to further advance epidemiologic surveillance of sCJD. Although there are many advantages of RT–QuIC, it is clear that this still has certain limitations. Firstly, for RT–QuIC, the recombinant substrate is very important. Therefore, the substrate needs to be standardized and manufactured on a large scale. Second, RT–QuIC has been successful in CSF as well as peripheral tissues. In the near future, there should be more studies on tissues with minimal intervention and that can be checked many times without affecting the patient. Since, as we know, RT–QuIC helps to diagnose at an early stage, minimal intervention therapy is of utmost importance. Thirdly, in prion diseases, RT–QuIC is a part of diagnostic standards. In the future, we should establish RT–QuIC assay as a diagnostic criterion in synucleinopathies and tauopathies.

## 4. RT–QuIC in Degenerative Neurological Diseases

Neurodegenerative diseases are complex, multifactorial, and heterogeneous diseases that are identified by the accumulation of misfolded proteins in the brain. Many neurodegenerative diseases are associated with the accumulation of specific misfolded proteins.

### 4.1. Human Prion Disease

RT–QuIC was first successful in the diagnosis of human prion disease. It is highly accurate in identifying misfolded forms of prion proteins from the CSF of patients with CJD and has already been included in the diagnostic criteria of sporadic CJD, the most common human prion disease (Table 1). The largest study analyzed CSF of 174 patients with gTSEs for 14-3-3, tau protein, S100b, and neuron-specific enolase (NSE). Biomarkers were found positive in 81% of gCJD and 69% in gTSE, while most of the GSS and FFI patients were negative for 14-3-3 and t-tau [31]. A study in 56 CSF samples including 20 cases of GSS with P102L mutation, 12 cases of fatal familial insomnia (FFI), and 24 cases of genetic CJD (gCJD) detected prion protein in CSF samples by RT–QUIC. Western blotting was used to analyze 14-3-3 protein, and t-tau protein was measured using an ELISA. They found the sensitivities of CSF RT–QUIC 78%GSS, 100% FFI, and 87% gCJD E200K, and 100% V203I, and all samples had the specific was 100%. Furthermore, the sensitivities of biomarkers were lowered: 11% in GSS, 0% in FFI, and 73% in gCJD E200K 67% in V203I [32].

Using these first-generation assays, RT–QuIC was shown to have a sensitivity of 77–89% and a specificity around 100% [9,33,34,35,36]. The specificity of first-generation RT–QuIC assays is close to 100% and is significantly better than that of surrogate markers such as CSF 14-3-3 or tau protein [34].

The presence of PrPSc in olfactory mucosa (OM)has been mentioned by many previous studies. Table 1 shows that there are successful studies using RT–QuIC to detect prion seeding activity in OM [28,35]. One study showed that RT–QuIC response in OM tissue is faster than CSF, with a 97% sensitivity and a 100% specificity [35]. Another study performed RT–QuIC on CSF and OM samples from 86 patients. The results showed that the sensitivity was 86% for CSF and 97% for OM, and the specificity was 100% [28].

**Table 1 pathogens-10-00305-t001:** Real-time quaking-induced conversion (RT–QuIC) for prion disease.

	Number	Number (Controls)	Sensitivity	Specificity	Samples	Reference
definite cases of sCJD	34	165	85	100	CSF	[9]
definite cases of sCJD	123	103	89	99	CSF	[33]
genetic prion disease	56	50	83	100	CSF	[32]
definite and probable cases of sCJD	15	43	77	100	CSF	[35]
	13		97	100	OM	
definite cases of sCJD	10	1	FG: 91 SG: 96	100	CSF	[27]
definite and probable cases of sCJD	43	100	77	100	CSF	[36]
definite cases of sCJD	81	64	FG: 69 SG:94	100	Skin samples	[29]
definite and probable cases of sCJD	276	915	81	94	CSF	[34]
genetic prion diseases	17		91			
variable protease-sensitive prionopathy	1		0			
definite cases of sCJD	12	15	100	100	Skin	[37]
sCJD	10	17	SG: 97	100	OM	[28]
	12		FG: 72 SG: 86	100	CSF	
definite cases of sCJD	174	82	SG: 92–95%	98.5–100	CSF	[24]
definite cases of sCJD	11	6	100	100	Eyes	[38]
	65	118	97	99	CSF	[39]
definite cases of sCJD	12	2	100	100	Peripheral nerve	[40]
sCJD	4	10	100	100	Digestive organs	[41]
all CJD patients	32	37	Ha23–231: 68.6	100	Skin punch biopsies	[42]
all CJD patients	102	80	96	100	CSF and OM	[43]

A previous study showed that second-generation RT–QuIC helped to increase the diagnositic sensitivity and speed. These results were confirmed in a large study [29], which showed that the second-generation RT–QuIC assay had better reproducibility. The RT–QuIC assay detected the prion seeding activity in skin samples of sCJD with a sensitivity of 69% in first-generation RT–QuIC but higher in second-generation RT–QuIC at 94%. Another study showed that the sensitivity of RT–QuIC assay in CSF was 72% for first-generation RT–QuIC and 86% for second-generation RT–QuIC. Moreover, RT–QuIC was also successful for diagnosis using other samples such as peripheral nerves [40] and digestive organs, with the SD_50_ of digestive tissues reaching >10^6–8^/g [41].

### 4.2. Synucleinopathies

PD, multiple system atrophy (MSA), dementia with Lewy bodies (DLB), or Lewy body dementia, are synucleinopathies caused by the abnormal accumulation of protein aggregates called α-synuclein in the brain. Outside prion RT–QuIC assays, α-synuclein RT–QuIC is the most developed RT–QuIC (α-synuclein RT–QuIC) seed amplification platform to aid in premortem diagnosis, largely using CSF specimens.

Despite being developed in the last five years, αSyn RT–QuIC assays have been adapted for a range of diagnostically relevant human biospecimens, including the brain and CSF, olfactory mucosa, submandibular gland, and skin. (Table 2)

It has previously been shown that α-synuclein can be detected in the brain and biological fluids such as CSF and serum [44,45,46,47,48,49,50,51,52,53,54,55].

**Table 2 pathogens-10-00305-t002:** Real-time quaking-induced conversion (RT–QuIC) for the synucleinopathy.

Disease	Sample	Number	Number (Controls)	Sensitivity	Specificity	Reference
DLB and PD	Brain and CSF	12	20	92	100	[44]
	CSF	17	15	65	100	
		20		95		
PD	Brain and CSF	76	ND	88.5	94	[45]
DLB		40		100		
MSA		10		89		
MSA and PD	Brain	7	2	100	100	[46]
DLB	DLB brain tissues	17	28	94	100	[47]
PD		12		92		
PB, MSA	OM	18	18	81.8	84.4	[48]
CBD, PSP		11		16.7		
DLB	CSF	29	49	93	96	[49]
MSA		1		100	100	
PD, incidental Lewy body	Submandibular gland tissues (FFPE)	15	11	100	100	[50]
LB	CSF	21	101	95	84–98	[54]
DLB		7		94		
PD		34		97		
iRBD		28		93		
PAF		18		100		
PD and other synucleinopathies.	Skin	57	73	93–94	93–98	[56]

The first successful use of α-syn RT–QuIC was applied to detect α-synuclein aggregation in the brain and CSF from dementia with LB and PD patients. The results gave the sensitivities of 92% and 95%, with an overall specificity of 100%. Patients with tauopathies showed negative results. In particular, they found that three iRBD patients also had a positive RT–QuIC response, suggesting that the method could be used as an early diagnostic of synucleinopathy [44].

A comparably more rapid α-syn RT–QuIC was developed, which demonstrated assay times of 1–2 days [45] compared to the 5–13 days shown in a previous study. The study assay also demonstrated 93% sensitivity and 100% specificity with blinded analysis of 29 synucleinopathy cases (12 PD and 17 DLB) and controls, including AD [47].

RT–QuIC assay application to the largest cohort of CSF samples has been examined. The method accurately detected α-synuclein seeding activity, including DLB, PD, iRBD, and PAF, with a sensitivity of 95.3%. In contrast, in samples of MSA patients, the assays gave no α-synuclein seeding activity, showing that MSA and LBD are associated with different conformational strains of α-synuclein. The study also showed 98% specificity in a neuropathological cohort of 101 cases lacking LB pathology [54].

A recent study applied the RT–QuIC assays to detect CSF α-synuclein with the isolated rapid-eye-movement sleep behavior disorder. With 52 isolate REM sleep behavior disorder (iRBD) and 9 controls, the results showed that the sensitivity was 90% and the specificity was 78%. During follow-up, 32 (61.5%) patients were diagnosed with PD or DLB 3.4 ± 2.6 years after LP, and 31 (96.9%) of these were positive for CSF α-synuclein in the RT–QuIC, while none of the controls developed synucleinopathy [55].

Some studies using the RT–QuIC assays to detect α-syn on peripheral tissues [48,50,56,57]. With the α-synuclein in brushes of OM, α-synuclein RT–QuIC showed 55.5% and 81.8%, positive in PD and MSA, respectively. In contrast, only 16.7% were positive in CBD and PSP. Finally, the assay showed 84.4% specificity. The most recent research has been successful in detecting phosphorylated α-synuclein (PαSyn) seeding activity in the skin by RT–QuIC [56,57]. RT–QuIC analysis of the αSynP seeding activity in autopsy abdominal skin samples revealed a 93–94% sensitivity and a 93–98% specificity in synucleinopathies (PD, DLB, and MSA) [56].

### 4.3. Tauopathies

Diseases involving tau pathology include those with preferential aggregation of 3 microtubule binding-repeat (3R), 4-repeat (4R), or both 3R and 4R (3R/4R) tau isoforms. The predominant 3R tau isoforms can be seen in In Pick disease (PiD). The 3R and 4R isoforms are deposited in AD. 4R tauopathies include corticobasal degeneration (CBD), progressive supranuclear palsy (PSP), globular glial tauopathy (GGT), and argyrophilic grain disease (AGD).

It has previously been shown that the Aβ and tau PMCA/RT–QuIC may work in diagnosing AD by distinguising between AD patients and other neurodegenerative disorders or controls by protein misfolding cyclic amplification assay (Ab-PMCA) with more than 90% of sensitivity and specificity using cerebrospinal fluid [58]. AD RT–QuIC assay can detect tau filaments in AD and chronic traumatic encephalopathy (CTE) from the brain tissue. Moreover, Aβ misfolding and aggregation is thought to precede tau misfolding and aggregation, although previous studies have shown amazing results in detecting AD even at the preclinical stage by biomarkers (CSF Aβ42/40, t-tau, and p181-tau) [59]. However, the most useful of the tau-RT–QuIC will be the differential diagnoses of primary tauopathies (Table 3).

The main success of tau RT–QuIC assay based on the seeded polymerization is that it can detect and distinguish PiD from other tauopathies, neurological diseases, and healthy controls. This method could detect the tau of seeding activity in 2 µL aliquots of PiD brain dilutions down to 10^−7^–10^−9^; PiD seeding activities were 100-fold higher in the frontal and temporal lobes than in the cerebellar cortex [60].

Further developments in tau assay have resulted from the advent of AD RT–QuIC, which is the first RT–QuIC 3R/4R tau test to detect tau fibers in AD and chronic traumatic encephalopathy (CTE) from brain tissue, known as AD RT–QuIC [61]. The method can detect as little as 16 fg of pure synthetic tau fibrils. It has been shown that the tau seeding of AD and CTE are significantly different from PiD as well as different types of tau diseases such as those with 4R tau aggregation [61].

In the latest study, a group of a tau RT–QuIC for 4R tauopathies have been developed, specifically PSP, CBD, and FTDP-17 MAPT. This assay could detect disease-associated subtypes of 4R tau seeds as indicated by differences in their conformational templating of the fibrillar tau products of 4R RT–QuIC. Moreover, this study showed that the tau RT–QuIC assay can detect tau seeding activity in CSF collected from living patients. 4R RT–QuIC can detect 4R seeds in the brain tissue with up to 10^3^–10^6^-fold sensitivity. 4R RT–QuIC analysis of postmortem CSF of the PSP and CBD CSFs each had sensitivities of 100%, while 0/7-unaffected controls were positive, giving a diagnostic specificity of 100%. However, 4R RT–QuIC analysis of CSF had 69% and 50% of positive reactions for PSP and CBD/CBS, respectively [64].

Although only a short development, tau RT–QuIC has also had considerable success in diagnosing different types of taupathies with high sensitivity and specificity (Table 3).

## 5. Conclusions

Over the last 10 years, the RT–QuIC assay has gained significant attention among the available laboratory aids for the clinical diagnosis of neurodegenerative disorders. Although there are still certain limitations, RT–QuIC has become the most effective tool for the in vivo diagnosis of not only prion diseases but also synucleinopathies and tauopathies. Moreover, the advances in RT–QuIC also help direct valuable research for the early diagnosis and treatment of most of these devastating disorders.

## Figures and Tables

**Table 3 pathogens-10-00305-t003:** Real-time quaking-induced conversion (RT–QuIC) for the tauopathies.

Sample	Number and Disease	Result	Reference
Brain and CSF	8 cases: PiD 13 cases: AD	Tau RT–QuIC that can detect tau seeds in 2 µl aliquots of PiD brain dilutions down to 10^−7^–10^−9^. PiD seeding activities were 10^2^-fold higher in frontal and temporal lobes compared to cerebellar cortex. Strikingly, this test was 10^3^- to 10^5^-fold less responsive when seeded with brain containing predominant 4-repeat (4R) tau aggregates.	[60]
	7 cases: PSP, 4 cases: CBD, 3 cases: FTDP-17		
Brains	16 cases: AD	AD RT–QuIC detected seeding activity in AD brains at dilutions as extreme as 10^7^–10^10^-fold but was 10^2^–10^6^-fold less responsive when seeded with brain from most cases of other types of tauopathy	[61]
Brains	11 cases: AD, 4 cases: PiD, 3 cases: PSP, 2 cases: FTLD 2 cases: control subjects	Using full-length recombinant tau substrates to detect tau seeding activity in AD and other human tauopathies, it will contribute to the further development of early detection of AD and other tauopathies	[62]
Brains	8 cases: 3R Tau 13 cases: 3R/4R Tau 13 cases: 4R Tau	K12 RT–QuIC assay allows the ultrasensitive detection and discrimination of both 3R and 3R/4R types of pathological tau using a single tau substrate (K12CFh)	[63]
Brain and CSF	+ 4R tau pathology included 16 cases: PSP, 9 cases: CBD, 3 cases: FTDP- 17 *MAPT* with the P301L mutation, 5 cases: FTDP17 *MAPT* with the N279K mutation and IVS10 + 3G > A mutation, and 3R predominant 4R tau deposition. + 3R/4R tau pathology from 6 sporadic AD, 3 cases: familial AD, 3 cases: control subjects, and 3 cases: PART. + 3R Tau from 8 cases PiD	Developed 4R RT–QuIC for the 4-repeat (4R) tau aggregates of PSP, CBD, and other diseases with 4R tauopathy. The assay detected seeds in 10^6^–10^9^-fold dilutions of 4R tauopathy brain tissue but was orders of magnitude less responsive to brain with other types of tauopathy, such as from AD.	[64]

## Data Availability

No new data were created or analyzed in this study. Data sharing is not applicable to this article.

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
