# Peer review of "The Latest Research on RT-QuIC Assays—A Literature Review"

_pathogens, 2021, doi:10.3390/pathogens10030305_

Round 1
Reviewer 1 Report
In this submission the authors wrote a review about RT-QuIC and how it would be useful as a tool for diagnosing different neurodegenerative diseases related with the aggregation of PrP, α-synuclein and tau proteins. The review would be useful and interesting to read for scientist interested in this topic, particularly to catch up with the latest results of this technique.
However, it has some major issues that should be fixed previous to publication.
1-English language and style should be reviewed. There are some sentence that has no sense. Other are not connected with the rest of the text. Two example would be:
Line 121: "RT-QuIC protocols to differentiate between sCJD subtypes and detect atypical prion 121 diseases with higher sensitivity in vivo." These sentence is not connected to the rest of the text. Maybe it has moved from other part of the text accidentally.
Line 140: "...QuIC kinetics and age at onset of disease, disease duration of CSF sampling, or timing of the lumbar puncture..." What are the authors referring with "duration of CSF sampling"? They refer to the time the CSF samples has been stored? The state of the patient when the sample was taken?
2-Section 4.4: Recently, it has been plubished an article about the use of different salts in RT-QuIC for the detection of PrP, synuclein and tau aggregates and how these salts improve the technique (DOI: 10.1073/pnas.1909322116). I think the authors should add some comment about these results.
There are other minor issues to amend before publication:
1-Line 41: The authors give data of AD worldwide prevalence from 2010. It would be better to use more up-to-date data.
2-Line 62-63: The authors should indicate that proteinase K is the most used protease for the discrimination of PrPC and PrPSc.
3-Line 79: The authors say: "PMCA could have tremendous implications for research and diagnosis of vCJD." However, it would be more fair to say that PMCA have been pivotal to confirm, for example, the protein-only hypothesis itself, by generating an infectious prion using only recombinant PrP expressed in E. coli and some cofactors (DOI of reference: 10.1126/science.1183748). I know that there is no direct relation between this comment and the diagnosis of TSEs, but this technique has been tremendously useful in the last decade and it should be noted in the review.
3-Lines 83-86: The authors say: "Moreover, although RT-QuIC is as sensitive as PMCA in the detection of as little as 1 fg PrPSc in the brains of sporadic CJD (sCJD) patients, it is much quicker than PMCA, with results being obtained within 2 days rather than 4–5 days." This information is not correct, as a PMCA assay could be performed with reasonable reliance in 2-3 days. I think the authors should center in other disadvantages of PMCA that makes this technique highly unsuitable for the diagnosis of TSEs in hospitals or other medical centers. For example, it requires brain homogenate, the sonication setup is much more difficult and expensive to maintain, it requires PK digestion of samples and western blotting for the detection of resistant PrP, etc. I think these are much more important disadvantages that makes the RT-QuIC more attractive for the diagnosis of TSEs. In my opinion, the authors should talk extensively about this matter.
4-Lines 104-105: The authors say: "Another major goal of this study was to determine whether eQuIC can also detect prion seeds naturally occurring in the blood of humans infected with vCJD." The authors should indicate the result of this experiments or eliminate this sentence.
5-Lines 119-120: "There is also a need for improvements in the diagnosis of prion diseases." It would be interesting that the authors give their opinion about some improvements that would be tested and have not been done yet.
6-It would be interesting to add some comments about the diagnosis of other TSEs (Fatal Familiar insomnia or Gerstmann-Straüssler-Scheinker disease) using RT-QuIC. There is an article regarding the use of Bank Vole recombinant PrP to detect prions from different TSEs that would be referenced (DOI: 10.1371/journal.ppat.1004983).
7-Line 150: The authors say: "A cut-off value of <1250 × 106/L has been...". It should be changed for : "A cut-off value of <1250 × 106 red blood cells per liter has been...".
8-Table 1-There is no indication about what the authors are referring with the acronyms FG or SG that appears in the table. It should be noted in the caption of the table.
9-In the discussion, it would be interesting to compare with more details the specificity and sensitivity of RT-QuIC vs other diagnostic methods (14-3-3, neurofilament light chain, etc.).
Author Response
15th February 2021
Dr. Lawrence S. Young
Editor-in-Chief
Pathogens
Dear Dr. Lawrence S. Young,
We wish to re-submit the manuscript titled “The latest research on RT-QuIC assays: A literature review” to Pathogens. The manuscript ID is pathogens-1106301 - Revision.
We thank you and the reviewers for your thoughtful suggestions and insights. The manuscript has benefited from these insightful comments. We have made various changes to the text in accordance with the provided feedback. We look forward to working with you and the reviewers to move this manuscript closer to publication in Pathogens.
According to the reviewer feedback, the manuscript has been rechecked and the necessary changes have been made to address the reviewers’ suggestions. The responses to all comments have been prepared and attached herewith.
Thank you for your consideration. I look forward to hearing from you.
Sincerely,
Katsuya Satoh
Department of Locomotive Rehabilitation Science,
Nagasaki University Graduate School of Biomedical Sciences
satoh-prion@nagasaki-u.ac.jp
Tel.: +81-958-19-7991
Review 1:
The manuscript was significantly revised as pointed out by reviewer 1.
In addition, we wrote comments to address the concerns of the reviewer.
>1-English language and style should be reviewed. There are some sentence that has no sense. >Other are not connected with the rest of the text. Two example would be:
Line 121: "RT-QuIC protocols to differentiate between sCJD subtypes and detect atypical prion 121 diseases with higher sensitivity in vivo." These sentence is not connected to the rest of the text. Maybe it has moved from other part of the text accidentally.
Line 140: "...QuIC kinetics and age at onset of disease, disease duration of CSF sampling, or timing of the lumbar puncture..." What are the authors referring with "duration of CSF sampling"? They refer to the time the CSF samples has been stored? The state of the patient when the sample was taken?
>2-Section 4.4: Recently, it has been plubished an article about the use of different salts in RT->QuIC for the detection of PrP, synuclein and tau aggregates and how these salts improve the >technique (DOI: 10.1073/pnas.1909322116). I think the authors should add some comment >about these results.
Thank you very much for your comments. We have corrected the inappropriate sentences.
There are other minor issues to amend before publication:
1-Line 41: The authors give data of AD worldwide prevalence from 2010. It would be better to use more up-to-date data.
We have added content to lines 41-43. at 1st submitted manuscript.
(In dementia, AD is most common, with a worldwide prevalence of approximately 35 million people in 2010 [2,3] and increasing 50 million people in 2020 [4]. In 2020, total payments for Alzheimer’s and other dementias will amount to $305 billion [5].)
2-Line 62-63: The authors should indicate that proteinase K is the most used protease for the discrimination of PrPC and PrPSc.
Thank you for pointing this out. We have explained this in lines 65-69.
Resistance to proteinase K (PK) digestion is considered a definition of the transmissible isoform of the prion protein (PrP(Sc)). Thus, the gold standard for the diagnosis of prion disease is the detection of proteinase K-resistant prion protein (PrPSc). Like amyloid fibers found in Alzheimer's and other amyloid-related disorders or tau aggregation in tauopathies, PrPSc is a variable polymer that can aggregate large fibers.
>3-Line 79: The authors say: "PMCA could have tremendous implications for research and >diagnosis of vCJD." However, it would be more fair to say that PMCA have been pivotal to >confirm, for example, the protein-only hypothesis itself, by generating an infectious prion >using only recombinant PrP expressed in E. coli and some cofactors (DOI of reference: >10.1126/science.1183748). I know that there is no direct relation between this comment and >the diagnosis of TSEs, but this technique has been tremendously useful in the last decade and >it should be noted in the review.
We recognize this limitation and have now explained the content in lines 86-93.
PMCA can be applied in many fields and has great significance in research and diagnosis. This technique is used to understand the biology of unique prions to detect ultrasensitive prions and to screen for molecules that interfere with the formation of prions. Furthermore, the prions produced by PMCA can be infected in animals of various wild-type species. This suggests that the infectious agent in TSEs is composed exclusively of protein. Previous studies have also shown the infection material was created by PMCA using brain extracts, cell lysates, high purity PrPSc, and even recombinant PrP produced in bacteria [22].
>4-Lines 83-86: The authors say: "Moreover, although RT-QuIC is as sensitive as PMCA in the >detection of as little as 1 fg PrPSc in the brains of sporadic CJD (sCJD) patients, it is much >quicker than PMCA, with results being obtained within 2 days rather than 4–5 days." This >information is not correct, as a PMCA assay could be performed with reasonable reliance in 2->3 days. I think the authors should center in other disadvantages of PMCA that makes this >technique highly unsuitable for the diagnosis of TSEs in hospitals or other medical centers. For >example, it requires brain homogenate, the sonication setup is much more difficult and >expensive to maintain, it requires PK digestion of samples and western blotting for the >detection of resistant PrP, etc. I think these are much more important disadvantages that makes >the RT-QuIC more attractive for the diagnosis of TSEs. In my opinion, the authors should talk >extensively about this matter.
Thank you for the comment, we have used the old document, therefore the new content has not been updated. We have added the following content in lines 97-106.
RT-QuIC is as sensitive as PMCA in the detection of as little as 1 fg PrPSc in the brains of sporadic CJD (sCJD) patients [9,23]. However, PMCA requires brain homogenate, the sonication setup is much more difficult and expensive to maintain, and it requires PK digestion of samples and western blotting for the detection of resistant PrP. This makes this technique highly unsuitable for the diagnosis of TSEs in hospitals or other medical centers. In contrast, RT-QuIC uses a substrate called rPrP, which is reproducible. In addition, the RT-QuIC reaction products are not contagious and therefore do not require specialized containment laboratories. Finally, RT-QuIC, besides being low-cost and based on a multiwall Thioflavin T (ThT) detection form, is better adapted for use as a high-throughput diagnostic assay than PMCA.
>5-Lines 104-105: The authors say: "Another major goal of this study was to determine whether >eQuIC can also detect prion seeds naturally occurring in the blood of humans infected with >vCJD." The authors should indicate the result of this experiments or eliminate this sentence.
We have indicated the result in lines 116-121.
Therefore, an adaptation of RT-QuIC, called e-QuIC, was developed [26]. A study showed that the eQuIC assay can detect up to 1014-fold dilutions and uses less than ~1 ag of PrPRes vCJD brain homogenates in human plasma, making it more sensitive than the previous study by about 104-fold. However, the assay was difficult to use to standardize and less reproducible than the standard RT-QuIC (given that the eQuIC is significantly more sensitive than the RT-QuIC).
>6-Lines 119-120: "There is also a need for improvements in the diagnosis of prion diseases." It >would be interesting that the authors give their opinion about some improvements that would >be tested and have not been done yet.
We have added our opinion to lines 131-140 at 1st submitted manuscript..
Despite the many advantages of RT-QuIC, it is clear that there remain limitations. Firstly, for the RT-QuIC, the recombinant substrate is very important. Therefore, the substrate needs to be standardized and manufactured on a large scale. Second, RT-QuIC has been successful in CSF as well as peripheral tissues. In the near future, there should be more studies on tissues with minimal intervention, which can be checked many times without affecting the patient. As RT-QuIC aids early diagnosis, minimal intervention therapy is of utmost importance. Third, in prion diseases, RT-QuIC forms part of core diagnostic standards. In the future, we should establish an RT-QuIC assay as a diagnostic criterion in synucleinopathies and tauopathies.
>7-It would be interesting to add some comments about the diagnosis of other TSEs (Fatal >Familiar insomnia or Gerstmann-Straüssler-Scheinker disease) using RT-QuIC. There is an >article regarding the use of Bank Vole recombinant PrP to detect prions from different TSEs >that would be referenced (DOI: 10.1371/journal.ppat.1004983).
We have added a comment on the diagnosis of other TSEs (Fatal Familiar insomnia or Gerstmann-Straüssler-Scheinker disease) using RT-QuIC in Line 152-159.
A study on 56 CSF samples including those from 20 cases of GSS with P102L mutation, 12 cases of fatal familial insomnia (FFI), and 24 cases of genetic CJD (gCJD) detected prion protein in CSF samples using RT-QUIC, analyzed 14-3-3 protein using western blotting, and measured t-tau protein using ELISA. They found that the specificity to be 100% and sensitivities of CSF RT-QUIC to be 78% with GSS, 100% with FFI, 87% with gCJD E200K, and 100% with V203I. Besides, the sensitivities of biomarkers were lowered: 11% in GSS, 0% in FFI, and 73% in gCJD E200K 67% in V203I [32].
>8-Line 150: The authors say: "A cut-off value of <1250 × 106/L has been...". It should be >changed for : "A cut-off value of <1250 × 106 red blood cells per liter has been...".
Thank you very much.
>9-Table 1-There is no indication about what the authors are referring with the acronyms FG or >SG that appears in the table. It should be noted in the caption of the table.
We have explained the abbreviation.
>10-In the discussion, it would be interesting to compare with more details the specificity and >sensitivity of RT-QuIC vs other diagnostic methods (14-3-3, neurofilament light chain, etc.).
We have added a comment to lines 149-158.
The largest study analyzed CSF of 174 patients with gTSEs for 14-3-3, tau protein, S100b, and neuron-specific enolase (NSE). Biomarkers were found to be positive in 81% of gCJD and 69% in gTSE, while most of the GSS and FFI patients were negative for 14-3-3 and t-tau [31]. A study on 56 CSF samples including those from 20 cases of GSS with P102L mutation, 12 cases of fatal familial insomnia (FFI), and 24 cases of genetic CJD (gCJD) detected prion protein in CSF samples using RT-QUIC, analyzed 14-3-3 protein using western blotting, and measured t-tau protein using ELISA. They found that the specificity to be 100% and sensitivities of CSF RT-QUIC to be 78% with GSS, 100% with FFI, 87% with gCJD E200K, and 100% with V203I. Besides, the sensitivities of biomarkers were lowered: 11% in GSS, 0% in FFI, and 73% in gCJD E200K 67% in V203I [32].

Reviewer 2 Report
In their work, Trang Đồng and Katsuya Satoh summarize the most recent developments of RT-QuIC and PMCA techniques for the detection of misfolded PrP, α-syn and tau proteins in different biological matrices. The authors focused more on RT-QuIC and PMCA applications for misfolded prions (due to their background, probably) and provided less information regarding RT-QuIC and PMCA applications on synucleinopathies and tauopathies. The RT-QuIC and PMCA techniques are nowadays more than similar, at least for α-syn. The respective inventors of α-syn RT-QuIC and α-syn PMCA decided to address these techniques as seeding aggregation assays (SAA)1.Ultrasensitive protein misfolding detection assays are a hot topic, the tables provided by the authors are useful in summarizing the reviewed studies, but there are several points to be improved:
line 40 – the period of the sentence “Prion–like mechanisms in AD is the most common type of tauopathy [2]” is nonsense, it seems a copy and paste typo. Moreover, AD is commonly thought to be a secondary tauopathy. Aβ misfolding and aggregation is thought to precede tau misfolding and aggregation. Although Aβ and tau PMCA/RT-QuIC may work in diagnosing AD (an example of PMCA for Aβ can be found here2), it is not worthy to invest effort on that since CSF biomarkers (e.g. CSF Aβ42/40, t-tau and p181-tau) work amazingly in detecting AD3 even at the preclinical stage. Thus, I suggest to highlight that the tau-RT-QuIC will be mostly useful for the differential diagnosis of primary tauopathies, like PSP and CBD.
Line 54 – have tau-RT-QuIC results been compared to CSF t-tau and p-tau levels? If yes, it will be interesting to discuss about it.
Line 126 – “4. Factors affecting RT-QuIC” these factors are discussed only for PrP RT-QuIC, please discuss them in the review or make the review purpose clearer. Another option could be deleting this chapter to improve the balance between PrP RT-QuIC and all the other applications.
Line 164 – “Furthermore, a pH of 7.4 provided more rapid RT-QuIC reactions than more acidic pH values.” for example, for α-syn RT-QuIC the opposite is true, more acidic pH values favor the aggregation while more basic pH values inhibit it.
Line 198 – remove the dot in “α-. synucleinopathies"
Lines 204-206 – better discuss the rapid growth of SAA for a-synuclein by pointing out that synucleinopathies lack of specific and reliable markers (like for AD) and, in the case of PD, the slow disease progression gives chances for therapeutic intervention. α-syn SAA, up to now, seems the best diagnostic strategy4,5.
Table 2 – there is a reference missing for RT-QuIC on skin biopsies6.
Lines 210-212 – Revise the period, as it is written it seems that a-syn RT-QuIC is able to differentiate AD from controls.
Lines 222-223 – “A recent study applied the RT-QuIC developed by Fairfoul et al. [42]. The assay yielded positive results in 47 (90.4%) patients and two (22.2%) controls.” Please specify the type of patients, the innovative part of this study is that they applied the protocol to iRBD patients by following them longitudinally, to see how many of them would develop clinical synucleinopathies.
Some suggested references
- Kang, U. J. et al. Comparative study of cerebrospinal fluid α-synuclein seeding aggregation assays for diagnosis of Parkinson’s disease. Movement Disorders 34, 536–544 (2019).
- Salvadores, N., Shahnawaz, M., Scarpini, E., Tagliavini, F. & Soto, C. Detection of misfolded Aβ oligomers for sensitive biochemical diagnosis of Alzheimer’s disease. Cell Rep 7, 261–268 (2014).
- Jack, C. R. et al. NIA-AA Research Framework: Toward a biological definition of Alzheimer’s disease. Alzheimers Dement 14, 535–562 (2018).
- Gaetani, L. et al. CSF and Blood Biomarkers in Neuroinflammatory and Neurodegenerative Diseases: Implications for Treatment. Trends Pharmacol Sci 41, 1023–1037 (2020).
- Manne, S. et al. Blinded RT-QuIC Analysis of α-Synuclein Biomarker in Skin Tissue From Parkinson’s Disease Patients. Movement Disorders 35, 2230–2239 (2020).
Author Response
15th February 2021
Dr. Lawrence S. Young
Editor-in-Chief
Pathogens
Dear Dr. Lawrence S. Young,
We wish to re-submit the manuscript titled “The latest research on RT-QuIC assays: A literature review” to Pathogens. The manuscript ID is pathogens-1106301 - Revision.
We thank you and the reviewers for your thoughtful suggestions and insights. The manuscript has benefited from these insightful comments. We have made various changes to the text in accordance with the provided feedback. We look forward to working with you and the reviewers to move this manuscript closer to publication in Pathogens.
According to the reviewer feedback, the manuscript has been rechecked and the necessary changes have been made to address the reviewers’ suggestions. The responses to all comments have been prepared and attached herewith.
Thank you for your consideration. I look forward to hearing from you.
Sincerely,
Katsuya Satoh
Department of Locomotive Rehabilitation Science,
Nagasaki University Graduate School of Biomedical Sciences
satoh-prion@nagasaki-u.ac.jp
Tel.: +81-958-19-7991
Review 2
The manuscript was significantly revised according to the comments of reviewer 2.
In addition, we have also written comments to address the feedback of reviewer 2.
>1. line 40 – the period of the sentence “Prion–like mechanisms in AD is the most common type of >tauopathy [2]” is nonsense, it seems a copy and paste typo. Moreover, AD is commonly thought to >be a secondary tauopathy. Aβ misfolding and aggregation is thought to precede tau misfolding and >aggregation. Although Aβ and tau PMCA/RT-QuIC may work in diagnosing AD (an example of >PMCA for Aβ can be found here2), it is not worthy to invest effort on that since CSF biomarkers > (e.g. CSF Aβ42/40, t-tau and p181-tau) work amazingly in detecting AD3 even at the preclinical >stage. Thus, I suggest to highlight that the tau-RT-QuIC will be mostly useful for the differential >diagnosis of primary tauopathies, like PSP and CBD.
We rewrote the content in lines 233-242 at 1st submitted manuscript.
It was previously shown that the Aβ and tau PMCA/RT-QuIC may serve to diagnose AD by distinguishing between AD patients with another neurodegenerative disorder and control by protein misfolding cyclic amplification assay (Ab-PMCA) with more than 90% of sensitivity and specificity using cerebrospinal fluid [58]. An AD RT-QuIC assay can detect tau filaments in AD and chronic traumatic encephalopathy (CTE) from brain tissue. Moreover, Aβ misfolding and aggregation is thought to precede tau misfolding and aggregation. Previous studies have shown very amazing results in detecting AD even at the preclinical stage by biomarkers (CSF Aβ42/40, t-tau, and p181-tau)[59]. However, the most useful aspect of the tau-RT-QuIC will be in the differential diagnoses of primary tauopathies.
>2.Line 54 – have tau-RT-QuIC results been compared to CSF t-tau and p-tau levels? If yes, it will >be interesting to discuss about it.
Thank you for the comment. Unfortunately, there is no manuscript to compare the CSF t-tau, p-tau levels and tau-RT-QuIC assay results with.
>3.Line 126 – “4. Factors affecting RT-QuIC” these factors are discussed only for PrP RT-QuIC, >please discuss them in the review or make the review purpose clearer. Another option could be >deleting this chapter to improve the balance between PrP RT-QuIC and all the other applications.
We agree with the comment and have removed this section accordingly. Thank you very much.
>4. Line 164 – “Furthermore, a pH of 7.4 provided more rapid RT-QuIC reactions than more acidic >pH values.” for example, for α-syn RT-QuIC the opposite is true, more acidic pH values favor the >aggregation while more basic pH values inhibit it.
Thank you for the feedback. We found your comment to be reasonable and have therefore removed this section. Thank you very much.
>5. Line 198 – remove the dot in “α-. synucleinopathies"
Thank you for the feedback. We rewrote this.
>6. Lines 204-206 – better discuss the rapid growth of SAA for a-synuclein by pointing out that >synucleinopathies lack of specific and reliable markers (like for AD) and, in the case of PD, the >slow disease progression gives chances for therapeutic intervention. α-syn SAA, up to now, seems > the best diagnostic strategy4,5.
A recent paper reported the establishment of a high-sensitivity ELISA system for phosphorylated and oligomer α-synuclein. No similar treatise has been published since now. However, quantitative results cannot be established by the RT-QUIC method in cerebrospinal fluid examinations. There are advantages and disadvantages to all measurement methods.
>7.Table 2 – there is a reference missing for RT-QuIC on skin biopsies6.
There are two groups of authors who researched RT-QuIC on skin biopsies and we have added this to the manuscript. Wang, Z., Becker, K., Donadio, V., Siedlak, S., Yuan, J., Rezaee, M., et al. Skin α-Synuclein Aggregation Seeding Activity as a Novel Biomarker for Parkinson Disease. JAMA neurology 2020, 78(1), 1–11. Advance online publication. https://doi.org/10.1001/jamaneurol.2020.3311.
Manne, S. et al. Blinded RT-QuIC Analysis of α-Synuclein Biomarker in Skin Tissue From Parkinson’s Disease Patients. Movement Disorders 35, 2230–2239 (2020).
>8.Lines 210-212 – Revise the period, as it is written it seems that a-syn RT-QuIC is able to
>differentiate AD from controls.
We rewrote this in lines 196-201 at 1st submitted manuscript.
>9.Lines 222-223 – “A recent study applied the RT-QuIC developed by Fairfoul et al. [42]. The >assay yielded positive results in 47 (90.4%) patients and two (22.2%) controls.” Please specify the >type of patients, the innovative part of this study is that they applied the protocol to iRBD patients >by following them longitudinally, to see how many of them would develop clinical >synucleinopathies.
We rewrote the content in lines 206-211 at 1st submitted manuscript
>10. RT-QuIC assay application to the largest cohort of CSF samples has been examined. The >method accurately detected α-synuclein seeding activity including DLB, PD, iRBD, and PAF, >with the sensitivity of 95.3%. In contrast, in samples of MSA patients the assays gave no α->synuclein seeding activity, it is shown that MSA and LBD are associated with different >conformational strains of α-synuclein. And then, the study also gave that 98% specificity in a >neuropathological cohort of 101 cases lacking LB pathology [54].
Thank you for these references. This was very useful.
Some suggested references
Kang, U. J. et al. Comparative study of cerebrospinal fluid α-synuclein seeding aggregation assays for diagnosis of Parkinson’s disease. Movement Disorders 34, 536–544 (2019).
Salvadores, N., Shahnawaz, M., Scarpini, E., Tagliavini, F. & Soto, C. Detection of misfolded Aβ oligomers for sensitive biochemical diagnosis of Alzheimer’s disease. Cell Rep 7, 261–268 (2014).
Jack, C. R. et al. NIA-AA Research Framework: Toward a biological definition of Alzheimer’s disease. Alzheimers Dement 14, 535–562 (2018).
Gaetani, L. et al. CSF and Blood Biomarkers in Neuroinflammatory and Neurodegenerative Diseases: Implications for Treatment. Trends Pharmacol Sci 41, 1023–1037 (2020).
Manne, S. et al. Blinded RT-QuIC Analysis of α-Synuclein Biomarker in Skin Tissue From Parkinson’s Disease Patients. Movement Disorders 35, 2230–2239 (2020).
Round 2
Reviewer 1 Report
The authors have responded to all the suggested comments, improving the manuscript. However, some minor english changes should be done to improve the final manuscript previous to publication.
Author Response
Reviewer 1
The authors have responded to all the suggested comments, improving the manuscript. However, some minor english changes should be done to improve the final manuscript previous to publication.
Reviewer 2 Report
The revised version of the manuscript addressed most of the points arisen. However, In my opinion, there is still some English editing needed. For example:
line 41 - "In the dementia, AD is the most common" I would change it to "Among dementias, AD is the most common"
line 55 - "it has become a useful tool use of protein synthesis" I did not catch the meaning of this.
I recommend a general language revision of the whole manuscript.
Author Response
Reviewer 2
The revised version of the manuscript addressed most of the points arisen. However, In my opinion, there is still some English editing needed. For example:
line 41 - "In the dementia, AD is the most common" I would change it to "Among dementias, AD is the most common"
I have rewritten at line 41
Among dementias, AD is the most common…
line 55 - "it has become a useful tool use of protein synthesis" I did not catch the meaning of this.
I have rewritten at line 56.
After more than a decade of the advent and development of the RT-QuIC assays [8,9], it has become a useful tool in the laboratory to aid in the clinical diagnosing of neurodegenerative disorders.
I recommend a general language revision of the whole manuscript.
Thank you very much for your comments
We had been checked and corrected the errors in English to complete the manuscript.